# MONTAGE-AGNOSTIC AND CALIBRATION-FREE EEG EVENT SEGMENTATION

## ABSTRACT

Accurate segmentation and detection of events in continuous electroencephalography (EEG) signals are critical for advancing brain-computer interfaces (BCIs) and understanding neural dynamics, but generalization across diverse recording montages and datasets remains a major challenge. Traditional methods for EEG analysis have largely relied on manual segmentation or time-locked experiments, which struggle with real-world scenarios that involve irregularly timed stimuli triggering or endogeneous events from the BCI operator. Although progress has been made in segmentation tasks such as sleep staging, those approaches primarily address longer, well-defined segments and do not generalize to short-duration stimuli events, such as events typically used in BCI control. In this work, we introduce a novel montage-agnostic framework with spatial interpolation of learned channel embeddings. This approach enables high temporal resolution segmentation in continuous EEG that operates without training on the target recording montage, requiring no subject-specific calibration. Validated across diverse datasets and different paradigms spanning P300, SSVEP and motor imagery, our model consistently outperforms the original foundation models including BIOT and EEGPT, demonstrating superior cross-dataset and montage-agnostic generalization. By removing montage dependency, our framework lays the groundwork for future real-time applications in neuroscience research, clinical diagnostics, and the development of brain-computer interfaces. The code and pre-processed datasets are available at: https://anonymous.4open.science/r/BN00FFgN2H1mhYhVC10J-4505

Table 1: Comparison of EEG analysis methods on whether they can: **Cross-Dataset:** Cross-Dataset generalization. **Montage-Agnostic:** Electrode-montage agnostic. **Spatial:** Spatial Information awareness. **HighRes:** High temporal resolution. **Calibration-Free:** Calibration-free transfer to unseen montages and datasets.

| Method | Cross-Dataset | Montage-Agnostic | Spatial | HighRes | Calibration-Free |
|---|---|---|---|---|---|
| EEGNet (Lawhern et al., 2018) | ✗ | ✗ | ✗ | ✓ | ✗ |
| BIOT (Yang et al., 2023) | ✓ | ✗ | ✗ | ✓ | ✗ |
| EEGPT (Wang et al., 2024) | ✓ | ✗ | ✓ | ✓ | ✗ |
| EEG-GNN (Demir et al., 2021) | ✗ | ✗ | ✓ | ✓ | ✗ |
| U-Time (Perslev et al., 2019) | ✓ | ✗ | ✗ | ✗ | ✗ |
| Ours | ✓ | ✓ | ✓ | ✓ | ✓ |

## 1 INTRODUCTION

Electroencephalography (EEG) is the most widely used non-invasive, portable neuro-imaging technique, offering millisecond temporal resolution but suffering from low signal-to-noise ratios and susceptibility to motion and electrophysiological artefacts. Among the many EEG applications, including sleep staging, motor imagery and clinical monitoring, paradigms such as P300 and steady-state visually evoked potential (SSVEP) exploit this high temporal acuity to study rapid cortical responses to discrete stimuli, providing the basis for cognition-centred brain-computer interfaces (BCIs). Accurate analysis in these paradigms has traditionally required precise synchronization

between stimulus events and the recorded EEG, limiting deployments to laboratory environments where event timing is controllable, while in real world scenarios stimuli occurrences are unpredictable and precise timings are unavailable. Most existing algorithms further assume fixed montages or specific acquisition hardware, limiting their ability to generalize across different EEG systems and setups.

This work investigates whether recent *foundation models* for EEG can lift the synchronization requirement by segmenting continuous EEG signals and detecting event-related activity without requiring task-specific fine-tuning. We formulate the problem of event detection in continuous EEG as a high-resolution segmentation task that assigns a label to every 4 ms window of continuous EEG, to indicate whether it contains a stimulus-related event or not. Unlike existing EEG segmentation tasks such as sleep staging, which operate on very low temporal resolutions (e.g., one label every 30 seconds), our approach achieves significantly higher temporal precision. Two research questions therefore arise: (i) Can large pre-trained representations be effectively applied to continuous EEG data to automatically segment and detect events without time-locked information? (ii) How does our approach to event detection in continuous EEG on unseen datasets and montages compare to traditional, time-locked methods across diverse BCI tasks?

We address these challenges by introducing a lightweight pre-processing layer that learns a mapping from arbitrary input channel electrode locations, enabling the use of frozen foundation model backbones. A shallow segmentation head is attached to different encoders, EEGPT (Wang et al., 2024), BIOT (Yang et al., 2023), and the convolutional baseline EEGNet (Lawhern et al., 2018), and the resulting systems are evaluated on eight public corpora covering P300, SSVEP and motor imagery, selected from the Mother of All BCI Benchmarks (MOABB) (Aristimunha et al., 2024) and a Rapid Serial Visual Presentation (RSVP) dataset (Davis et al., 2022).

In summary, we make the following contributions:

- A lightweight, plug-in pre-processing module that enables montage-agnostic inference with frozen EEG backbones.
- A systematic evaluation of foundation, transformer and CNN models on asynchronous EEG segmentation at 4 ms resolution.
- A framework for calibration-free, cross-montage event detection in continuous EEG, approaching the performance of traditional time-locked methods.
- An open-source implementation of the proposed method, together with the pre-trained weights and evaluation scripts released for reproducibility.

## 2 RELATED WORK

**Time-Locked EEG Analysis and BCI Limitations**   Traditional EEG decoding relies on event-related potentials (ERPs) like the P300 (Farwell & Donchin, 1988) and SSVEP, requiring precise time-locked averaging to overcome low signal-to-noise ratios. This synchronization bottleneck confines BCIs to controlled lab settings (Gehring et al., 1993), as real-world scenarios lack precise event markers. While methods like Common Spatial Patterns (CSP) improved motor imagery classification (Pfurtscheller & Da Silva, 1999), they required per-subject calibration and failed to generalize across sessions (Ang et al., 2008), hindering plug-and-play applications.

Early deep learning approaches demonstrated that CNNs could learn spatio-temporal features directly from raw EEG (Schirrmeister et al., 2017). EEGNet (Lawhern et al., 2018) advanced this by emulating band-pass filtering and CSP through depthwise convolutions, achieving cross-paradigm generalization. However, these models still assumed fixed electrode montages and required full retraining for new tasks, which is a critical limitation for real-world deployment with heterogeneous devices.

**Approaches to Montage Heterogeneity**   Many previous work attempted to address the challenge of montage heterogeneity through interpolation, such as synthesizing signals for missing channels using Riemannian domain adaptation (Mellot et al., 2024) or using graph neural networks to explicitly model spatial relationships (Han et al., 2023), or employing dynamic spatial attention (Guetschel et al., 2024). While effective, these methods often require access to target data for adaptation or are evaluated in cross-subject settings within a single dataset (Ouahidi et al., 2023), rather than a

cross-dataset setting without any calibration. In contrast, our work introduces a method that modifies a frozen foundation model, utilizing its learned channel embeddings to enable montage-agnostic inference on unseen datasets without calibration on target dataset.

**Self-Supervised Foundation Models** Recent work utilizes self-supervised masked pre-training to learn universal representations from large, unlabeled datasets. BENDR (Kostas et al., 2021) follows wav2vec2 (Baevski et al., 2020) with masked autoencoding, while BIOT (Yang et al., 2023) tokenizes multichannel biosignals. EEGPT (Wang et al., 2024) scales further with a variety of EEG datasets across paradigms using spatio-temporal alignment with channel embeddings. This area is rapidly evolving, with recent models like CSBrain (Zhou et al., 2025), BrainOmni (Xiao et al., 2025), REVE (El Ouahidi et al., 2025), and LUNA (Döner et al., 2025) pushing the boundaries of scale and performance, or focusing calibration-free transfer (Grizou et al., 2025; Zhang et al., 2025). While many of these new models solve montage heterogeneity by training massive, monolithic architectures from scratch, our work explores an alternative, parameter-efficient direction. We focus on modifying existing, frozen foundation models, which remains a practical and important challenge for broader application of these foundation models.

**Calibration-free EEG Segmentation** While cross-dataset classification has been shown for pre-segmented trials (e.g., motor imagery (Duan et al., 2020), SSVEP (Wang et al., 2023)), continuous event detection remains underexplored. Existing segmentation focuses on low-resolution tasks like sleep staging (Perslev et al., 2019), where one label every 30 seconds is annotated by human expert. No prior work evaluates foundation models for high-temporal-resolution segmentation at 4ms level across paradigms.

Prior cross-dataset transfers often failed due to electrode mismatch. Solutions like EEGNet (Lawhern et al., 2018) and EEG-GNN (Demir et al., 2021) assumed same montages across datasets. BIOT (Yang et al., 2023) and U-Time (Perslev et al., 2019) used fixed montages, and EEGPT (Wang et al., 2024) assumed standardized 10-20 channel labels but allowed absence of channels. In various datasets where electrodes are placed differently, data from these electrodes are either ignored or mapped to the same channel, leading to loss of spatial information and degradation of model performance. Therefore, applying existing models to high-resolution segmentation on arbitrary montages without any calibration remains an open challenge.

**Challenges in EEG Segmentation.** EEG segmentation is a challenging task, primarily explored in sleep staging applications. Previous work has focused on classifying sleep stages based on manually annotated labels, which are used to train models like U-Time (Perslev et al., 2019) and SleepEEGNet (Mousavi et al., 2019). These tasks operate at low temporal resolutions (e.g., one label every 30 seconds) and rely on manually annotated labels derived from multiple sensors. In contrast, event-related EEG paradigms, such as SSVEP and P300, require high temporal precision, with events lasting less than 500 ms. Detecting such events in continuous EEG recordings poses two significant challenges:

**Temporal Resolution:** High temporal precision is required to capture rapid neural responses, which is challenging due to EEG's low signal-to-noise ratio (SNR). Traditional ERP analysis relies on synchronized, time-locked data epochs to average out noise, but this approach is impractical for detecting events directly from continuous recordings.

**Annotation Difficulty:** Unlike sleep staging, where labels are manually curated, ERP analysis depends on precise synchronization between stimuli and recorded EEG. However, delays between stimuli presentation and brain responses introduce variability that cannot be accurately labeled. This reliance on synchronization further limits the applicability of traditional methods to real-world scenarios.

Our work addresses these challenges by a parameter-efficient montage-agnostic layer that unlocks frozen foundation models for unseen EEG systems. We utilize the existing event markers in the datasets to train a segmentation model on top of several frozen backbones, and evaluate the model on unseen datasets without any time-locked information. Consequently, our framework provides the first demonstration of using a parameter-efficient adapter to enable a frozen foundation model for montage-agnostic and cross-dataset generalization for high-resolution EEG event detection without time-locked information.

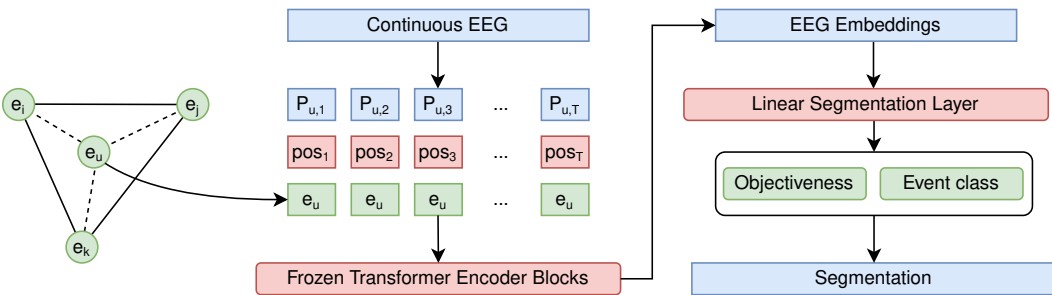

Figure 1: The proposed architecture model and the segmentation head. The channel embeddings of unseen channels are interpolated from known channels based on physical coordinates, then added to EEG patch embeddings. The frozen foundation model computes montage-agnostic embeddings, and the linear segmentation layer predicts a label block. Finally the predicted blocks are voted to obtain segmentation on continuous EEG of arbitrary length.

## 3 METHOD

Our pipeline augments a frozen foundation encoder with two light-weight modules: (i) a **montage-agnostic preprocessing layer** that converts any electrode layout into the canonical token space expected by the encoder, and (ii) a **segmentation head** that predicts high-resolution event labels. We employ the publicly available EEGPT weights (Wang et al., 2024), but the design is agnostic to the particular backbone, as long as it accepts a sequence of EEG tokens with channel embeddings. We first recap its relevant mechanics and architecture, then describe our proposed preprocessing pipeline, and finally detail the segmentation head and training objective for the task of high temporal resolution segmentation in continuous EEG data.

### 3.1 EEGPT TOKENIZATION AND CHANNEL EMBEDDINGS

EEGPT uses a local spatio-temporal embedding method to patch and embed the EEG signal. Let the vectorized EEG data be $X \in \mathbb{R}^{C \times T}$ of $C$ channels and $T$ samples, where $X_{i,j}$ is the $j$-th sample of the $i$-th channel. Each channel is sliced into non-overlapping segments of length $l$ samples, then linearly projected into a latent space of dimension $d$ as $P_{i,j} \in \mathbb{R}^d$, where $P_{i,j}$ is the $j$-th patch of the $i$-th channel. Each channel is assigned a learnable embedding vector $e_i \in \mathbb{R}^d$ of the same dimension as the latent space, which is learned jointly with the model parameters during masked pre-training. The channel embeddings are then added to the patch embeddings, in addition to the standard rotary positional embeddings $pos_j$ (Su et al., 2024), before being fed into the transformer encoder. The resulting token $t_{i,j}$ for the $i$-th channel and $j$-th time point is:

$$t_{i,j} = P_{i,j} + e_i + pos_j \tag{1}$$

### 3.2 MONTAGE-AGNOSTIC PREPROCESSING LAYER

In EEGPT, the channel embeddings are mapped by the channel names, assuming the standard 10-20 system. For unknown channels or channels with different names, the model discards the data from these channels completely, which leads to loss of spatial information and degradation of model performance. To address this problem and extend the model to unseen channels, we propose a preprocessing pipeline that interpolate the channel embeddings based on the physical locations of the electrodes, and compute the channel embeddings from the learned weights of the model.

Let $\mathcal{R} = \{r_1, \ldots, r_N\}$ be the $N$ reference electrodes with 3-D coordinates $\mathbf{p}_r \in \mathbb{R}^3$ and learnt embeddings $\mathbf{e}_r \in \mathbb{R}^d$, where $d$ is the dimension of the latent space. For a new recording with montage $\mathcal{U} = \{u_1, \ldots, u_{N_{\text{new}}}\}$ and coordinates $\mathbf{p}_u$, we synthesize an embedding for every unseen channel $u \in \mathcal{U}$.

We interpolate the channel embedding $e_u$ as a weighted sum of the known channel embeddings $e_r$ based on the distance between the new channel and the known channels.

$$e_u = \sum_{r \in \mathcal{R}} w(r, u) e_r \tag{2}$$

where $w(r, u)$ is the interpolation weight for the known channel $r$ and the new channel $u$ and $\sum_{r \in \mathcal{R}} w(r, u) = 1$.

Intuitively, we expect the neighboring channels are weighted more than the far away channels. We additionally want the weights to be smooth and sparse, as the far away channels should have exponentially smaller weights. So we use a temperature parameter $\tau$ and softmax function over the inverse distance between the new channel and the known channels to compute the interpolation weights:

$$d(r, u) = \|\mathbf{p}_r - \mathbf{p}_u\|_2 \tag{3}$$

$$w(r, u) = \frac{\exp\big(-d(r, u)/\tau\big)}{\sum_{r' \in \mathcal{R}} \exp\big(-d(r', u)/\tau\big)} \tag{4}$$

where $\tau$ is a temperature parameter that controls the smoothness of the weights.

### 3.3 HIGH-RESOLUTION SEGMENTATION HEAD

For the task of high-resolution segmentation, we attach a lightweight segmentation head to the frozen foundation model.

**Task definition.** Given a continuous EEG stream, we emit a label every $\Delta t$ ms indicating NONE or one of $K$ stimulus classes. For an instant event at time $t$, we mark all $\Delta t$-ms windows in the range $[t, t + D]$ as the event class, where $D$ is the default duration of the event, e.g., 500ms for Rapid Serial Visual Presentation (RSVP) at 500ms stride. We mark the class of the event based on the stimulus type from its original paradigm. For example, the event class may be TARGET or NON-TARGET for P300. All other windows are marked as NONE.

**Architecture.** Suppose the foundation model outputs $T$ tokens $\mathbf{z}_t \in \mathbb{R}^{d_{\text{model}}}$, each of dimension $d_{\text{model}}$, we follow the downstream task design of EEGPT (Wang et al., 2024) and attach a linear layer to the frozen model, to project the tokens to $K + 1$ logits per time step, where $K$ is the number of classes.

However, the imbalance between the number of NONE and event class labels makes the model underfit the event class. To address this, we aggregate $B$ adjacent $\Delta t$-ms windows into a block, with two labels per block: (1) $o_t$ is objective label, indicating whether the block contains an event or not, and (2) $y_t$ is the class label of the first event in the block, if any. We then train the model to predict $o_t$ and $y_t$ for each block, with the weighted cross-entropy loss separately for the objective and class labels:

$$\mathcal{L}_{\text{seg}} = \mathcal{L}_{\text{CE}}(o_t, \hat{o}_t) + \lambda_{\text{c}} \, \mathcal{L}_{\text{CE}}(y_t, \hat{y}_t) \tag{5}$$

where $\hat{o}_t$ and $\hat{y}_t$ are the predicted logits for the objective and class labels, respectively, and $\lambda_{\text{c}}$ is a hyperparameter that controls the relative importance of the two losses.

**Non-maximum suppression.** The EEGPT model accepts an EEG signal of up to 2 seconds. To process continuous EEG of arbitrary length, we use a sliding window of 2 seconds with a stride of 60 ms, thus producing overlapping predictions of label blocks. The final predictions are obtained by majority voting at each time step, among all overlapping blocks.

## 4 EXPERIMENTS

### 4.1 DATASETS

The experiments were conducted on a diverse set of eight public EEG datasets from various BCI tasks, including motor imagery (MI), P300, and steady-state visually evoked potential (SSVEP) experiments. Table 2 summarizes their key statistics. All datasets were used as continuous EEG rather than pre-epoched trials.

Table 2: Datasets used in this study. The corpora span distinct paradigms, sampling rates, montages and subject pools, making cross-dataset generalization challenging.

| Name | Task | Ch. | Hz | Subj. | Total Length |
|---|---|---|---|---|---|
| Facecat (Davis et al., 2022) | SSVEP | 32 | 2000 | 31 | 20h 59m 37s |
| BI2013a (Congedo et al., 2011) | P300 | 16 | 128 | 25 | 5h 30m 0s |
| BI2014a (Korczowski et al., 2019a) | P300 | 16 | 512 | 64 | 12h 24m 51s |
| BI2014b (Korczowski et al., 2019b) | P300 | 32 | 512 | 38 | 2h 21m 45s |
| BNCI2014_001 (Tangermann et al., 2012) | MI | 22 | 250 | 9 | 11h 36m 29s |
| BNCI2014_009 (Aricò et al., 2014) | P300 | 16 | 256 | 10 | 1h 38m 1s |
| BNCI2015_001 (Faller et al., 2012) | MI | 13 | 512 | 12 | 16h 41m 22s |
| EPFLP300 (Hoffmann et al., 2008) | P300 | 32 | 2048 | 8 | 2h 59m 39s |

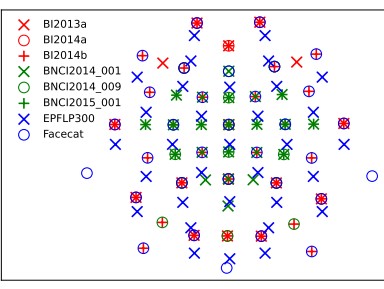

Figure 2: Electrode layouts for the eight corpora.

**Facecat Dataset**  The CVPR 2022 faces (Facecat) dataset (Davis et al., 2022) is a Rapid Serial Visual Presentation (RSVP) experiment where subjects are shown a series of images at fixed intervals of 500 ms. The dataset comprises 32-channel EEG recordings sampled at 2000 Hz from 31 subjects. The goal is to detect stimuli onset events and distinguish between relevant and irrelevant stimuli.

**MOABB Dataset Collection**  The Mother of All BCI Benchmarks (MOABB) collection (Aristimunha et al., 2024) includes datasets from a variety of EEG tasks, such as motor imagery (MI) and P300. The datasets vary in terms of the number of subjects, number of channels, and sampling rates. However, to evaluate our model performance, we need the raw continuous EEG data, instead of epoched data with time-locked pre-processing. We also require that the number of channels is not too small (at least 13 EEG channels) so that the topological maps are reliable.

## 4.2 Evaluation Metrics

We consider two types of evaluation metrics: classification metrics for the block classification model and segmentation metrics for the overall segmentation performance. In the classification setting, the model is to predict one label for each timestamp. Due to the nature of imbalance between the event and non-event classes, we choose to evaluate the model using the marco-F1 score. In the segmentation and event detection setting, however, we are interested in the temporal precision of the model in detecting the onset and duration of each event. To convert the classification results to segmentation results, we need to reconstruct the events from the predicted labels with non-maximum suppression. Alternatively, our block classification model can directly output the event onset and offset times, which can be used to evaluate the temporal precision of the model. The segmentation metrics include Intersection over Union (IoU) and mean distance between the predicted and true event onsets.

**Macro F1-Score.**  For each timestamp we predict one of $K+1$ labels: the $K$ event classes of the dataset and a *no-event* background class. We compute precision and recall for every class separately on the full timeline, compute the per-class F1 score, and obtain the macro F1-score by averaging the per-class F1 scores over all classes without weighting. For example, in the P300 task, each time point can be labeled as one of three classes: relevant stimulus, irrelevant stimulus, or non-stimuli. The F1-score is calculated for each of these classes separately and then averaged.

**Intersection over Union (IoU).**  For segmentation and event detection tasks, accurate label prediction does not necessarily imply accurate event detection. Instead, the Intersection over Union (IoU) metric is used to evaluate the overlap between the predicted and actual event segments. Suppose $A$ and $B$ are the predicted and ground truth segments, respectively, where $A$ is the set of timestamps predicted as an event, and $B$ is the set of timestamps labeled as an event. The IoU is calculated as $IoU = \frac{|A \cap B|}{|A \cup B|}$. For multi-class segmentation, the IoU is calculated for each class separately and then averaged.

## 4.3 Baseline Models

To benchmark the performance of the proposed models, several baseline models were used for comparison. We split the EEG data into fixed-length epochs with overlapping windows, then applied

Table 3: Results on single dataset with cross-subject evaluation. The best results are in **bold** and the second best are in underline.

| | | Random | BIOT | EEGNet | Original EEGPT | Proposed Method |
|---|---|---|---|---|---|---|
| BI2013a | F1 | $0.3366 \pm 0.0002$ | $0.4182 \pm 0.0148$ | $0.3840 \pm 0.0149$ | $0.4639 \pm 0.0013$ | **$0.4864 \pm 0.0195$** |
| | IoU | $0.2155 \pm 0.0001$ | $0.2983 \pm 0.0020$ | $0.2665 \pm 0.0077$ | **$0.3365 \pm 0.0072$** | $0.3315 \pm 0.0201$ |
| BI2014a | F1 | $0.3389 \pm 0.0001$ | $0.4123 \pm 0.0013$ | $0.4162 \pm 0.0309$ | $0.4421 \pm 0.0094$ | **$0.4910 \pm 0.0097$** |
| | IoU | $0.2176 \pm 0.0001$ | $0.2983 \pm 0.0009$ | $0.2947 \pm 0.0180$ | $0.3304 \pm 0.0106$ | **$0.3447 \pm 0.0093$** |
| BI2014b | F1 | $0.3367 \pm 0.0003$ | $0.4072 \pm 0.0023$ | $0.3314 \pm 0.0430$ | $0.4753 \pm 0.0053$ | **$0.5247 \pm 0.0158$** |
| | IoU | $0.2297 \pm 0.0002$ | $0.3019 \pm 0.0045$ | $0.2525 \pm 0.0292$ | $0.3592 \pm 0.0046$ | **$0.3811 \pm 0.0138$** |
| BNCI2014_001 | F1 | $0.2000 \pm 0.0001$ | $0.1951 \pm 0.0015$ | $0.2807 \pm 0.0248$ | $0.2884 \pm 0.0533$ | **$0.3044 \pm 0.0201$** |
| | IoU | $0.1829 \pm 0.0001$ | $0.1870 \pm 0.0006$ | $0.2312 \pm 0.0089$ | **$0.2406 \pm 0.0305$** | $0.2333 \pm 0.0207$ |
| BNCI2014_009 | F1 | $0.3334 \pm 0.0002$ | $0.3496 \pm 0.0162$ | $0.4341 \pm 0.0070$ | $0.5443 \pm 0.0535$ | **$0.5535 \pm 0.0485$** |
| | IoU | $0.2205 \pm 0.0001$ | $0.2521 \pm 0.0238$ | $0.3005 \pm 0.0135$ | **$0.4017 \pm 0.0534$** | $0.3947 \pm 0.0491$ |
| BNCI2015_001 | F1 | $0.3333 \pm 0.0001$ | $0.3282 \pm 0.0028$ | $0.3954 \pm 0.0349$ | $0.4339 \pm 0.0106$ | **$0.4556 \pm 0.0361$** |
| | IoU | $0.3115 \pm 0.0001$ | $0.3187 \pm 0.0009$ | $0.3498 \pm 0.0112$ | **$0.3783 \pm 0.0073$** | $0.3742 \pm 0.0269$ |
| EPFLP300 | F1 | $0.3335 \pm 0.0002$ | $0.4572 \pm 0.0117$ | $0.4709 \pm 0.0194$ | $0.5028 \pm 0.0045$ | **$0.5298 \pm 0.0118$** |
| | IoU | $0.2658 \pm 0.0001$ | $0.3670 \pm 0.0018$ | $0.3615 \pm 0.0114$ | **$0.3940 \pm 0.0023$** | $0.3869 \pm 0.0143$ |
| Facecat | F1 | $0.3357 \pm 0.0001$ | $0.4756 \pm 0.0017$ | $0.4928 \pm 0.0088$ | $0.5235 \pm 0.0368$ | **$0.5906 \pm 0.0144$** |
| | IoU | $0.2144 \pm 0.0001$ | $0.3721 \pm 0.0033$ | $0.3607 \pm 0.0178$ | $0.3963 \pm 0.0369$ | **$0.4401 \pm 0.0131$** |
| Mean | F1 | $0.3169 \pm 0.0465$ | $0.3804 \pm 0.0843$ | $0.4007 \pm 0.0704$ | $0.4593 \pm 0.0798$ | **$0.4920 \pm 0.0849$** |
| | IoU | $0.2339 \pm 0.0380$ | $0.2994 \pm 0.0567$ | $0.3022 \pm 0.0502$ | $0.3546 \pm 0.0563$ | **$0.3608 \pm 0.0619$** |

the models to classify each epoch and predict the center label. The predicted labels were then used to reconstruct the event onsets and durations with non-maximum suppression.

**Random Baseline**   Random baseline is a simple model that randomly shuffles the labels of the EEG data, or assigns random labels uniformly if the ratio of the labels is unknown. It indicates the performance of the model by chance, and serves as a lower bound for the evaluation metrics.

**EEGNet**   EEGNet is a compact convolutional neural network (CNN) architecture designed for EEG-based BCI applications. It has been shown to perform well across various EEG tasks, including motor imagery and P300 classification, with minimal computational overhead.

**BIOT**   BIOT is a transformer-based model designed for biosignals with varying number of channels and sampling rates. We used the BIOT model with the default pre-training and fine-tuning settings, and adapted it for classification tasks with overlapping windows.

**Original EEGPT**   EEGPT is a transformer-based model designed for processing EEG data. We used the EEGPT model with the default pre-training and fine-tuning settings, and adapted it for classification tasks with overlapping windows. The depth convolutional trainable layers are not used, as they cannot be applied to unseen new datasets with different montages.

**Direct Interpolation**   Direct interpolation with raw EEG signals is a baseline where we apply field interpolation to the raw EEG signals by minimum-norm projection to the full set of electrodes of EEGPT, using the MNE library (Gramfort et al., 2013).

## 4.4 EXPERIMENTAL SETUP

We conducted two series of experiments to evaluate the performance of the proposed models and compare them against the baseline models, each with a more generic scenario: (1) cross-participant scenario and (2) a calibration-free cross-dataset generalization scenario. Large baseline models such as EEGPT and BIOT were frozen during training, while the segmentation head and EEGNet was trained with a starting learning rate of $8e - 4$ for 100 epochs with AdamW optimizer. All experiments were conducted on a single Nvidia V100 GPU with training time of less than 48 hours for each.

**Cross-Participant Scenario.**   In the first scenario, we split the data from a single dataset by participant. This process was repeated in 3-fold cross-validation, where the data from each participant was used as the test set once. This scenario evaluates the model's ability to generalize to new subjects

Table 4: Results on calibration-free cross-dataset generalization. The best results are in **bold** and the second best are in underline.

| | | Random | BIOT | Original EEGPT | Direct Interpolation | Proposed Method |
|---|---|---|---|---|---|---|
| BI2013a | F1 | $0.3078 \pm 0.0001$ | $0.1709 \pm 0.0001$ | $0.4597 \pm 0.0110$ | $0.4506 \pm 0.0052$ | **$0.4849 \pm 0.0001$** |
| | IoU | $0.1859 \pm 0.0001$ | $0.1145 \pm 0.0005$ | **$0.3312 \pm 0.0071$** | $0.3269 \pm 0.0043$ | $0.3305 \pm 0.0003$ |
| BI2014a | F1 | $0.3027 \pm 0.0001$ | $0.2507 \pm 0.0337$ | $0.4253 \pm 0.0058$ | $0.3825 \pm 0.0083$ | **$0.4390 \pm 0.0009$** |
| | IoU | $0.1836 \pm 0.0001$ | $0.1736 \pm 0.0263$ | **$0.3063 \pm 0.0039$** | $0.2748 \pm 0.0067$ | $0.2992 \pm 0.0013$ |
| BI2014b | F1 | $0.2857 \pm 0.0001$ | $0.2171 \pm 0.0453$ | $0.4133 \pm 0.0092$ | **$0.4572 \pm 0.0127$** | $0.4341 \pm 0.0048$ |
| | IoU | $0.1740 \pm 0.0001$ | $0.1610 \pm 0.0527$ | $0.2762 \pm 0.0111$ | **$0.3363 \pm 0.0115$** | $0.2881 \pm 0.0047$ |
| BNCI2014_009 | F1 | $0.3018 \pm 0.0002$ | $0.1575 \pm 0.0209$ | $0.4220 \pm 0.0026$ | $0.3218 \pm 0.0067$ | **$0.4951 \pm 0.0008$** |
| | IoU | $0.1821 \pm 0.0002$ | $0.0972 \pm 0.0112$ | $0.3178 \pm 0.0037$ | $0.2472 \pm 0.0046$ | **$0.3492 \pm 0.0020$** |
| EPFLP300 | F1 | $0.2384 \pm 0.0002$ | $0.2163 \pm 0.2055$ | $0.3206 \pm 0.0063$ | $0.2278 \pm 0.0075$ | **$0.3964 \pm 0.0028$** |
| | IoU | $0.1484 \pm 0.0002$ | $0.1757 \pm 0.1701$ | $0.2482 \pm 0.0008$ | $0.1723 \pm 0.0068$ | **$0.2804 \pm 0.0058$** |
| Facecat | F1 | $0.3101 \pm 0.0001$ | $0.2346 \pm 0.0003$ | $0.4763 \pm 0.0028$ | $0.4659 \pm 0.0066$ | **$0.5202 \pm 0.0104$** |
| | IoU | $0.1871 \pm 0.0001$ | $0.1809 \pm 0.0003$ | $0.3515 \pm 0.0028$ | $0.3571 \pm 0.0075$ | **$0.3670 \pm 0.0108$** |
| Mean | F1 | $0.2536 \pm 0.0731$ | $0.2079 \pm 0.0935$ | $0.4195 \pm 0.0500$ | $0.3843 \pm 0.0332$ | **$0.4616 \pm 0.0424$** |
| | IoU | $0.1546 \pm 0.0443$ | $0.1505 \pm 0.0805$ | $0.3052 \pm 0.0348$ | $0.2858 \pm 0.0246$ | **$0.3191 \pm 0.0326$** |

that were not seen during training. But the model is still trained on the same task, with the same stimuli classes and the same EEG montage.

**Calibration-free Cross-Dataset Generalization Scenario.** To simulate a more challenging real-world setting, we trained a model on all but one dataset and evaluated it on the held-out dataset. This scenario tests the model's ability to generalize to unseen montages, sampling rates. In addition, the model is tested without any calibration on the target dataset, which is the most challenging scenario. EEGNet baseline is excluded in this scenario, as it requires training set and test set to have the same montage. We run all experiments with 3 random seeds, and report the mean metrics across runs.

## 4.5 RESULTS

**Cross-participant evaluation** Table 3 reports the mean F1 and IoU scores. Our montage-agnostic layer with EEGPT delivers the best average performance (F1=$0.4920\pm0.0849$, IoU=$0.3608\pm0.0619$), improving on the strongest baseline (original EEGPT) by **+7.1 %** and **+1.7 %** relative, respectively. The gain is particularly pronounced on datasets with more channels (EPFLP300, Facecat), demonstrating the effectiveness of incorporating all available channels into the model.

The second best model is the original EEGPT, which has much higher F1-score and IoU than the other baseline models. We observed that BIOT has worse performance than original EEGPT and EEGNet, which possibly indicates that the frequency domain features it learned are not suitable for the short time scale of the event detection task.

**Calibration-free Cross-dataset evaluation** Table 4 reports the results for calibration-free cross-dataset generalization. Our proposed method again achieves the highest mean scores (F1=$0.4616\pm0.0424$, IoU=$0.3191\pm0.0326$), surpassing original EEGPT by **+10.0 %** and **+4.6 %**, and direct interpolation by **+20.1 %** and **+11.7 %**, respectively, and significantly outperforming the random baseline. In every dataset, our method consistently outperforms the original EEGPT and BIOT models, demonstrating its robustness across datasets.

In addition, we observe that the dataset EPFLP300 is the most challenging for all models to generalize to, as it has an original sampling rate of 2048 Hz, which is much higher than the other datasets. It also exhibits a severe class imbalance, shown in the performance drop of the random baseline. Although all baseline models perform poorly on this dataset, our proposed method still achieves the best performance of F1=$0.3964\pm0.0028$ and IoU=$0.2804\pm0.0058$, outperforming the original EEGPT by **+23.6 %** and **+13.0 %**, respectively.

## 4.6 RECONSTRUCTED ERP PLOTS

With the predicted event onsets times from the calibration-free cross-dataset evaluation, we can reconstruct the event-related potentials (ERPs) for each class. As shown in Figure 3, the calibration-free ERP curve from predicted events clearly distinguished target and non-target events, and closely

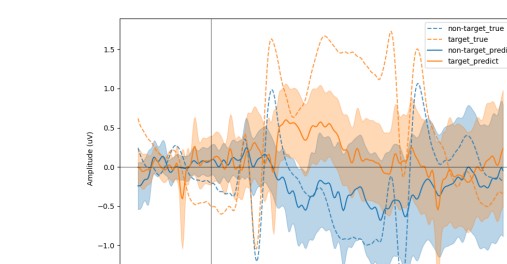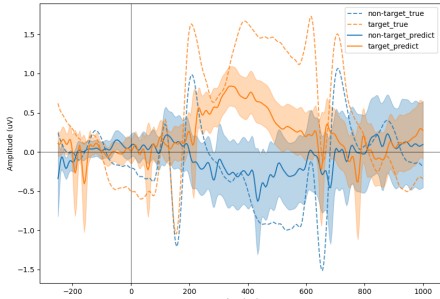

Figure 3: Calibration-free Event-related potentials (ERPs) reconstructed on the Facecat dataset Pz channel, from predicted events (solid) and ground truth events (dashed), Target (orange) and Non-target (blue) classes, with two different segmentation methods: EEGPT (left) and ours (right). The shaded area represents the 95 % confidence intervals across all epochs. The predicted ERP from our method closely resembles the ground truth ERP and shows P300 effect, demonstrating the model's ability to predict high temporal resolution events.

related to the ERP curve using ground truth event times. In addition, the P300 effect is observed at around 300ms after the stimulus onset. Although the peak is less pronounced than the ground truth curve, this observation further confirms the cross-dataset generalization ability of our model without any event timestamps or calibration. In contrast, the ERP curve from the original EEGPT model does not show any clear P300 effect, and its separation between target and non-target events is much weaker. More ERP plots are shown in the Appendix C.

## 5 CONCLUSION

In this work, we introduced a lightweight, parameter-efficient module that successfully enables montage-agnostic generalization with pre-trained, frozen EEG foundation models. By applying this module to the challenging task of high-resolution event segmentation, we demonstrated a practical path toward discarding the requirement for strict time-locked synchronization in many BCI paradigms, a key step in bridging the lab-to-field gap.

Our experiments directly answer our two research questions. First, we confirmed that large pre-trained representations can be effectively leveraged for high-resolution segmentation in continuous EEG. Second, we demonstrated that our proposed montage-agnostic layer is a viable and effective method for achieving robust cross-dataset generalization without any calibration on unseen datasets and montages. Our method based on interpolation of channel embeddings consistently outperformed both the original foundation models and traditional signal-space interpolation baselines, validating the benefits of learning a spatial mapping within the rich latent space of a pre-trained model.

With a parameter-efficient montage-agnostic layer and an augmented frozen EEG foundation model, we achieved high temporal resolution event segmentation and ERP-like event detection without any stimulus timestamps. The approach enables foundation EEG models with arbitrary montages, and achieves significantly better performance than all baselines in both cross-subject and calibration-free cross-dataset evaluations. The results indicate the potential of our approach to expand the range of EEG datasets that can be used for training and evaluation, and further improve the performance of EEG foundation models.

**Reproducibility Statement** We are committed to the reproducibility of our research. Our work relies exclusively on publicly available datasets (e.g., MOABB), and we provide detailed descriptions of our data preprocessing pipeline. For the review process, we have provided the full source code, and pre-processed data in an anonymous repository, as linked in the Abstract, at https://anonymous.4open.science/r/BN00FFgN2H1mhYhVC10J-4505. Upon publication, this repository will be formally released. The appendix includes key training and evaluation details.

**Ethics Statement** This research exclusively utilizes pre-existing, anonymized, publicly available datasets. Our work did not involve the collection of any new data from human subjects. The research that has been documented adheres to the ethical guidelines outlined by the ICLR.

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

## A   EXPERIMENTAL SETUP

All models are trained and evaluated on a single NVIDIA V100 GPU. Depending on the dataset size, the training time on a single dataset is from 4 to 20 hours. The evaluation time is less than 1 minute each. A batch size of 32 is used for all experiments.

## B   DETAILS FOR DATASETS

### BI2013A

BI2013a is a P300 dataset from a Brain Invaders experiment Congedo et al. (2011). The dataset has 16 Ag/AgCl electrodes (10-20 layout, reference = left ear-lobe) sampled at 512 Hz. The subject looks at a screen showing 6 by 6 grid of flashing aliens. A target alien is colored in cyan and non-target aliens are colored in light gray. During each repetition, a random group of aliens is selected to flash. The trial is labeled as a target if the target alien is flashing. A random interval between trials is used, with a mean of 100ms and in range [20, 500]ms following an exponential distribution.

### BI2014A

BI2014a is a P300 dataset from a Brain Invaders experiment Korczowski et al. (2019a). EEG was captured with 16 dry 8-pins gold-alloy electrodes g.Sahara (10-10 montage, reference = right earlobe) at 512 Hz. The subject looks at a screen showing 6 by 6 grid of aliens. One alien is colored in red as the target and the remaining are colored in gray as non-targets. 12 groups of 6 aliens are selected, with 2 groups including the target alien. During each repetition, each group of aliens is selected to flash in a pseudo-random order.

BI2014B

BI2014b is a multi-user P300 dataset from Brain Invaders Korczowski et al. (2019b). The dataset has 32 wet Ag/AgCl electrodes (10-10 layout, reference = right ear-lobe) sampled at 512 Hz. The experiment consisted of three conditions: each player playing solo and four sessions of playing with collaboration, in randomly session order. The subjects look at a screen showing 6 by 6 grid of aliens. One alien is colored in red as the target and the remaining are colored in gray as non-targets. 12 groups of 6 aliens are selected, with 2 groups including the target alien. During each repetition, each group of aliens is selected to flash in a pseudo-random order. Each align flashes eaxactly 2 times after all groups.

BNCI2014_001

BNCI2014_001 is a four-class motor-imagery dataset from the BCI competition IV 2a Tangermann et al. (2012). The dataset has 22 Ag/AgCl electrodes (reference = left mastoid) sampled at 250 Hz. The subject look at a screen. Each trial starts with a fixation cross and a warning tone at t = 0 s. At t = 2 s, a cue arrow pointing one of four directions (left, right, down, or up) appears and stays on the screen for 1.25 s, corresponding to one of the four classes (left hand, right hand, foot, or tongue). The subject is prompted to perform the motor imagery task until the fixation cross disappears at t = 6 s. No other feedback is provided. Each trial is labeled with one of the four motor-imagery classes.

BNCI2014_009

BNCI2014_009 is a P300 speller dataset from the BCI competition IV 2b Aricò et al. (2014). The dataset has 16 Ag/AgCl electrodes (10-10 layout, reference = right ear-lobe) sampled at 256 Hz. The subject looks at a screen showing 6 by 6 grid of letters. Before each trial, the subject is prompted to select a target letter. During each trial, one random row or column of letters is selected to intensify for 125ms, with an inter stimulus interval of 125ms. A trial is labeled as a target if the target letter is in the selected row or column.

BNCI2015_001

BNCI2015_001 is a two-class motor-imagery dataset Faller et al. (2012). The dataset has 13 Ag/AgCl electrodes (10-20 layout, reference = right ear-lobe) sampled at 512 Hz. The subject looks at a screen showing a green cross. At t = 3 s, an audio cue is played and a red arrow pointing down (both feet) or right (right hand) appears on the screen. Then a feedback bar appears on the screen between t = 4.25 s and t = 8 s. The subject is prompted to perform the motor imagery task until the feedback bar disappears at t = 8 s. A random 2-3 s pause is used between trials.

EPFLP300

EPFLP300 is a visual P300 dataset Hoffmann et al. (2008). The dataset has 32 electrodes (10-20 layout) sampled at 2048 Hz. The subject looks at a screen showing six images with household object (television, telephone, lamp, door, window, radio). Before each run, the subject is asked to count the number of times a target image (e.g., television) is flashed. During each run, in each block, and each trial, one of the six images is flashed for 100ms, followed by 300ms no image flashing. After the run, the subject is asked to report the number of times the target image was flashed to monitor the performance. A trial is labeled as a target if the target image is flashed.

FACECAT

The CVPR 2022 Faces (Facecat) dataset is a visual SSVEP dataset Davis et al. (2022). The dataset has 32 Ag/AgCl electrodes (10-20 layout) sampled at 2000 Hz. The subject looks at a screen showing a series of generated images of faces, with backgrounds masked with an elliptic grey frame. The stimuli images are grouped as one of eight classes: smile, no-smile, female, male, young, old, dark hair, or blond hair. Before each rapid serial visual presentation (RSVP) block for each class, the subject is asked to observe the images and take a mental note of seeing the target class. Each image


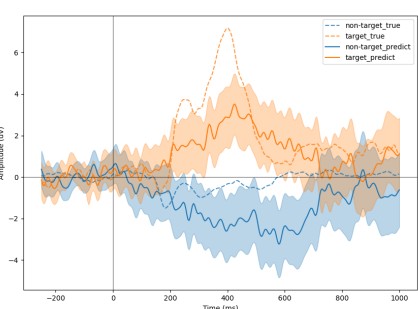
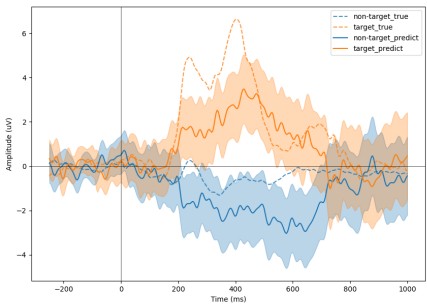

Figure 4: Calibration-free Event-related potentials (ERPs) reconstructed on the BI2014b test set, from predicted events (solid) and ground truth events (dashed), Target (orange) and Non-target (blue) classes, with two different channels: Pz (left) and Cz (right). The shaded area represents the 95 % confidence intervals across all epochs. The predicted ERP closely resembles the ground truth ERP, demonstrating the model's ability to predict high temporal resolution events.

is shown for 500ms without any pause between images. A trial is labeled as a target if the image belongs to the target class.

## C ADDITIOANL ERP PLOTS

## D LIMITATIONS

Despite the encouraging results, our study inherits several limitations that future work should address.

First, We uses a montage-agnostic layer to adapt EEGPT to new datasets with different montages, but the model still assumes an approximate correspondence between electrode labels and cortical areas within each dataset. Datasets collected with highly idiosyncratic caps, dry sensors, or extreme subject variability may therefore violate this assumption and degrade performance. With enough data, the model can learn to adapt to these variations, but it is expensive to collect large amounts of data for each subject.

Second, the backbone representations were learned from research-grade recordings with a large number of channels and high sampling rates, which may not generalize well to ultra-low-density consumer headsets. It is worth exploring whether the proposed method can be applied to EEG data with fewer channels and lower sampling rates, or even to other modalities such as ECoG or fNIRS.

## E LLM USAGE STATEMENT

Large Language Models (LLMs) were used as a writing assistant during the preparation of this manuscript. Specifically, they were only utilized for improving grammar, clarity, and latex formatting. The authors take full responsibility for the final content and claims made in this paper.

