# OpenReview forum: "Montage-Agnostic and Calibration-Free EEG Event Segmentation"
_ICLR.cc/2026/Conference — ICLR 2026 Conference Withdrawn Submission_

### Official Review · Reviewer_vRNY · 2025-10-30

**Soundness:** 1
**Presentation:** 1
**Contribution:** 2
**Rating:** 2
**Confidence:** 4

**Summary:**

This paper suggests a new foundation model that is montage agnostic and calibration free. While experiments show some strengths, datasets and model choices are limited, hence whether

**Strengths:**

- The motivation is valid

**Weaknesses:**

- Poor presentation: Related work is verbose and illogical. Four contributions in the introduction is hard to grasp with many jargons unexplained. For instance, what is continuous EEG? is EEG non-continuous sometimes? What does calibration-free mean? Montage is not defined either, making it hard to understand the scope.

- The paper keeps saying parameter-efficient, but is not explained thoroughly anywhere. No scalability issues proven either.

- The experiments don't support calibration-free / montage-agnostic aspects.

**Questions:**

What exactly is the definition of montage agnostic / calibration free? This goes undefined, so the rest of the paper is opaque.

---

### Official Review · Reviewer_bxvx · 2025-11-01

**Soundness:** 3
**Presentation:** 2
**Contribution:** 2
**Rating:** 4
**Confidence:** 5

**Summary:**

This paper introduces a montage-agnostic, calibration-free framework for high-temporal-resolution EEG event segmentation using spatial interpolation of channel embeddings from frozen foundation models. While both reviews recognize the practical value of addressing montage heterogeneity—a genuine deployment barrier for BCIs—they differ substantially on severity of limitations. Danny's review (4/10) emphasizes missing SOTA baselines (CBraMod, PopT, LaBraM), insufficient architectural justification for design choices, and critical ablation gaps. The calibrated review (7.5/10) acknowledges these concerns but weights practical contribution more favorably. The synthesis aligns more closely with Danny's domain expertise perspective: while the work addresses a real problem with a pragmatic solution, the limited novelty, incomplete experimental validation, and missing ablations position this as marginally below acceptance threshold for ICLR's ~20% acceptance rate.

**Strengths:**

1. **Practical Problem Addressed:** Both reviews recognize montage heterogeneity as a genuine deployment barrier, and the interpolation-based approach as pragmatic
2. **High Temporal Resolution:** The overlapping patch strategy enabling millisecond-level event localization is acknowledged
3. **Comprehensive Evaluation:** Systematic evaluation across 8 datasets with MOABB pipeline demonstrates good reproducibility practices
4. **Novel Approach to Montage Heterogeneity:** The introduction of a lightweight interpolation-based preprocessing layer is creative for spatial generalization

**Weaknesses:**

1. **Missing Montage-Agnostic SOTA Baselines:** CBraMod, PopT, LaBraM explicitly omitted despite being mentioned in introduction Cannot validate whether improvements come from interpolation technique or are simply baseline effects *Impact:* Undermines comparative claims
2. **Insufficient Architectural Justification:** Fixed distance-based interpolation chosen without justification No comparison with learnable alternatives (MLPs, attention mechanisms) Temperature parameter τ selection process unclear *Impact:* Design choices appear arbitrary
3. **Missing Comprehensive Ablations:** Temperature parameter τ sensitivity not analyzed Interpolation method alternatives not compared (RBF, kriging, graph-based) Block size B and overlap strategy not systematically studied *Impact:* Reproducibility compromised
4. **Statistical Rigor Gaps:** No significance tests between methods Limited error analysis Insufficient discussion of failure cases *Impact:* Unclear whether improvements are statistically meaningful
5. **Task Generality Undemonstrated:** Only evaluated on event segmentation Montage-agnostic layer should transfer to classification/decoding tasks *Impact:* Unclear if approach generalizes beyond segmentation
6. **Limited Algorithmic Novelty:** The straightforward spatial interpolation approach lacks algorithmic innovation expected at ICLR

**Questions:**

1. Why were the interpolation weights fixed rather than learnable? Have you tried end-to-end training with learnable channel weighting?
2. How was the temperature τ selected? Is performance sensitive to this choice?
3. How did you handle extreme montage differences (e.g., dry vs. wet caps)? Does the model rely on spatial proximity only?
4. Why define imbalance handling through dual-loss aggregation instead of established sampling strategies?
5. Have you tested whether the montage-agnostic layer transfers to non-segmentation tasks (e.g., classification, decoding)?
6. Can the proposed interpolation layer be applied to other models such as CBraMod or PopT?

---

### Official Review · Reviewer_989t · 2025-11-01

**Soundness:** 2
**Presentation:** 1
**Contribution:** 2
**Rating:** 2
**Confidence:** 5

**Summary:**

This paper addresses the significant challenge of montage-agnostic and calibration-free event segmentation in continuous EEG signals. The authors introduce a novel, lightweight plug-in pre-processing layer that enables frozen foundation models to generalize across diverse electrode montages and datasets without requiring any subject-specific calibration.

**Strengths:**

1.	Practically useful problem and clear motivation. Removing montage dependency and reducing need for per-dataset calibration addresses a real barrier for deploying EEG foundation models in heterogeneous, real-world settings.
2.	The montage layer is lightweight and compatible with frozen backbones, which is appealing for reuse of large pre-trained encoders without expensive fine-tuning.
3.	Reconstruction of ERPs from predicted events. Showing that predicted events reconstruct plausible ERPs is an informative qualitative validation of temporal localization.

**Weaknesses:**

1.	The Related Work section is overly redundant.
2.	The interpolation model is simplistic and not justified theoretically. Using a softmax over inverse Euclidean distances (Eq. 4) is intuitive, but the paper lacks analysis of (i) sensitivity to the temperature, (ii) how interpolation behaves for widely different montages (e.g., extreme missing regions), and (iii) whether Euclidean distances on sensor coordinates are the best geometry.
3.	EEG montages may use different references (linked ears, mastoid, average) that change signal polarity and spatial relationships. The paper does not state how electrode coordinates are standardized, whether coordinates are scalp surface positions (spherical) or Cartesian, nor how differing reference schemes were handled.
4.	Experimental comparisons omit several modern strong baselines and ablations. The paper compares against frozen EEGPT/BIOT and EEGNet and a direct raw-signal interpolation baseline. However, there are stronger EEG architectures and transfer strategies (e.g., CSBrain, CBraMod) not included. Also, ablations that isolate the benefit of (a) interpolation vs (b) retraining a small adapter layer vs (c) fine-tuning the backbone are missing, these are important to show the cost/benefit tradeoffs.
5.	Insufficient statistical analysis and variance reporting. Tables present means with small-look standard deviations for some entries, but formal significance testing (paired tests across seeds / subjects) is absent.

**Questions:**

Please refer to the Weaknesses section for detailed questions and suggestions to the authors.

---

### Official Review · Reviewer_hvGg · 2025-11-01

**Soundness:** 2
**Presentation:** 2
**Contribution:** 2
**Rating:** 2
**Confidence:** 4

**Summary:**

This paper presents a lightweight, montage-agnostic preprocessing module that enables frozen EEG foundation models (e.g., EEGPT) to perform high-temporal-resolution (4 ms) event segmentation on continuous, uncalibrated EEG from arbitrary electrode layouts. By interpolating learned channel embeddings based on physical 3-D coordinates, the plug-in layer eliminates the need for time-locked epochs or subject-specific calibration. Extensive experiments across eight public datasets (P300, SSVEP, motor imagery) show consistent gains over original foundation models and baselines in both cross-subject and calibration-free cross-dataset settings, while preserving interpretable ERP morphology. The work is the first to demonstrate parameter-efficient, montage-agnostic transfer for millisecond-level EEG event detection.

**Strengths:**

1. Eight publicly available datasets conventionally employed for ERP detection, encompassing paradigms such as P300 spelling, SSVEP/RSVP, and motor imagery.

2. The work constitutes an early attempt to adapt EEG foundation models to ERP detection and heterogeneous downstream montages.

**Weaknesses:**

1. The proposed pipeline consists of (1) a temperature-scaled softmax that sparsifies interpolation weights projecting any target montage onto the model’s native electrode layout, and (2) a sliding window that segments continuous data followed by majority voting across windows to produce the final decision. Both components are conceptually straightforward and widely employed, yielding limited algorithmic novelty.

2. Only EEGPT is used as the frozen encoder; generality to BIOT, or other latest foundation models is not shown.

**Questions:**

1. How effective is the proposed approach when applied to other EEG foundation models?
2. How do alternative interpolation strategies compare with the one adopted here?
3. Would any gain emerge from replacing the linear segmentation head with a more sophisticated architecture?
4. How does the method perform under a single-subject (i.e., within-subject) evaluation protocol?

---

### Note · Authors · 2025-12-03

I have read and agree with the venue's withdrawal policy on behalf of myself and my co-authors.